
# General properties of multiscalar RG Flows in $d = 4 - \varepsilon$

*Dedicated to the memory of Louis Michel (1923-1999), the first IHES professor of physics and a pioneer of group theory applications to RG flows.*

**Slava Rychkov[1,2] and Andreas Stergiou[3,4]**

**1** Institut des Hautes Études Scientifiques, Bures-sur-Yvette, France
**2** Laboratoire de physique théorique, Département de physique de l'ENS,
École normale supérieure, PSL University, Sorbonne Universités,
UPMC Univ. Paris 06, CNRS, 75005 Paris, France
**3** Theoretical Physics Department, CERN, 1211 Geneva 23, Switzerland
**4** Theoretical Division, MS B285, Los Alamos National Laboratory,
Los Alamos, NM 87545, USA

## Abstract

Fixed points of scalar field theories with quartic interactions in $d = 4 - \varepsilon$ dimensions are considered in full generality. For such theories it is known that there exists a scalar function $A$ of the couplings through which the leading-order beta-function can be expressed as a gradient. It is here proved that the fixed-point value of $A$ is bounded from below by a simple expression linear in the dimension of the vector order parameter, $N$. Saturation of the bound requires a marginal deformation, and is shown to arise when fixed points with the same global symmetry coincide in coupling space. Several general results about scalar CFTs are discussed, and a review of known fixed points is given.



# 1 Introduction

Renormalization group (RG) flows in scalar field theories have connections to innumerable problems in physics. One is usually interested in properties of these flows and their fixed points in physical dimension $d = 3$, and the classic approach to learn about such fixed points is to analytically continue from $d = 4 - \varepsilon$ dimensions [1]. In this paper we would like to consider the general case of the Wilson–Fisher RG equation for $N$ real scalar fields in $d = 4 - \varepsilon$ dimensions with the general quartic self-interaction

$$\tfrac{1}{4!} \lambda_{ijkl} \phi_i \phi_j \phi_k \phi_l \,, \tag{1}$$

with real symmetric tensor $\lambda_{ijkl}$. The one-loop beta-function has the well-known form[1]

$$\beta_{ijkl} = \frac{d}{dt} \lambda_{ijkl} = -\varepsilon \, \lambda_{ijkl} + B(\lambda_{ijmn} \lambda_{mnkl} + 2 \text{ permutations}) \,, \tag{2}$$

where $B = 1/16\pi^2$ in the standard normalization. From now on we will rescale the coupling so that $B = 1$ in the beta-function.

We will be studying the beta-function equation in the shown one-loop approximation. Ideally we would like to get a global picture of fixed points and RG flows described by this equation. Not much is actually known about this problem in full generality. As we will review below, a full classification of fixed points without any assumptions is available only for $N = 1, 2$. The problem of classifying fixed points can be seen as a difficult problem of real algebraic geometry.

In this paper our goal will be to review what is known about multiscalar fixed points, and to offer a few new general results which may guide future work towards full classification. Except for a few comments in section 6.3, we limit ourselves to the one-loop case as it is already sufficiently nontrivial.

In section 2 we analyze how stability of the quartic potential changes under RG flows. We show that fixed points have stable potential. We also show that while a stable potential may become unstable under RG flow, the inverse never happens.

In section 3 we give a representative review of many known classes of fixed points. This section also reviews a classic construction of fixed points with symmetries possessing one quadratic and two quartic invariants.

---

[1]Here $t = \ln(\mu/\mu_0)$, with $\mu$ the RG scale, is the RG time. In this paper we consider RG flows from UV to IR, i.e. $t$ is decreasing along the flow.

In section 4 we recall that the multiscalar RG flow is a gradient flow and present the corresponding height function, the *A*-function. Since the *A*-function decreases monotonically under RG flows, it's clearly of interest to know what is the minimal value it can take at a fixed point. We prove such a general bound, which scales linearly in $N$, in section 5. We show by examples that for almost all $N$, and in particular for all $N \geqslant 12$, our bound is best possible.

In section 6 we study RG stability of fixed points. We review general results about uniqueness of RG stable fixed points, and symmetry criteria for RG instability, following mostly the work of L. Michel. We also discuss and resolve a paradox of spurious zero eigenvalues of linearized RG equations.

Our main new result is the general bound on the *A*-function. We hope that this bound as well as our review of other existing general results will stimulate further work on general theory of multiscalar fixed points.

## 2   Stability of the potential

Physically, one is mostly interested in quartic couplings $\lambda_{ijkl}$ such that the potential is stable, which means that [2]

$$\lambda(\phi) \equiv \lambda_{ijkl}\,\phi_i\phi_j\phi_k\phi_l \geqslant 0 \qquad \text{for any real } \phi_i\,. \tag{3}$$

We will call $\lambda$ satisfying this condition "potential-stable" or simply "stable" (this should not be confused with RG-stability of RG fixed points, to be discussed in section 6).[2]

The set $\mathcal{C}$ of stable tensors $\lambda_{ijkl}$ forms a convex cone, which means that (a) if $\lambda \in \mathcal{C}$, then its rescaling $s\lambda \in \mathcal{C}$ for any $s \geqslant 0$, and (b) if two tensors $\lambda^{(1)}$ and $\lambda^{(2)}$ are in $\mathcal{C}$, then so are their convex linear combinations:

$$s\lambda^{(1)} + (1-s)\lambda^{(2)} \in \mathcal{C} \qquad \text{for any } 0 \leqslant s \leqslant 1\,. \tag{4}$$

We will refer to $\mathcal{C}$ as the stability cone.

We now wish to study RG trajectories which start in the complement of $\mathcal{C}$, so we pick a point $\lambda_{0,ijkl}$ in coupling space which is not in $\mathcal{C}$. This means that there exists a real $\bar{\phi}_i$ such that $\lambda_0(\bar{\phi}) < 0$. Let us do an infinitesimal RG flow step to the IR,

$$\lambda_0 \to \lambda = \lambda_0 + \Delta t\,\beta(\lambda_0), \quad \Delta t < 0\,, \tag{5}$$

and evaluate the quartic potential *on the same field configuration*, i.e. $\lambda(\bar{\phi})$. Using the form of the beta-function, we find that $\lambda(\phi)$ evolves according to

$$\frac{d}{d(-t)}\lambda(\bar{\phi}) = \varepsilon\,\lambda(\bar{\phi}) - 3V_{ij}V_{ij} \leqslant \varepsilon\,\lambda(\bar{\phi})\,, \tag{6}$$

where $V_{ij} = \lambda_{ijmn}\bar{\phi}_m\bar{\phi}_n$ and we used the fact $V_{ij}V_{ij} \geqslant 0$. From the form of this equation we see that if $\lambda_0(\bar{\phi}) < 0$ then the right-hand side is always negative and so the potential evaluated on the field configuration $\bar{\phi}$ gets more and more negative as the flow towards the IR progresses [2]. This has two consequences. First, the RG flow remains in the complement

---

[2]Condition 3 for four-tensors may be seen as a generalization of the condition for a symmetric matrix to be positive semidefinite. However it's quite more subtle than for matrices. For example checking this condition for a general four-tensor is NP-hard. Also, it's not true that a stable four-tensor can be written as a positive linear combination of elementary tensors $y_i y_j y_k y_l$ for different $y \in \mathbb{R}^N$ (even allowing for infinite combinations). See [3,4].

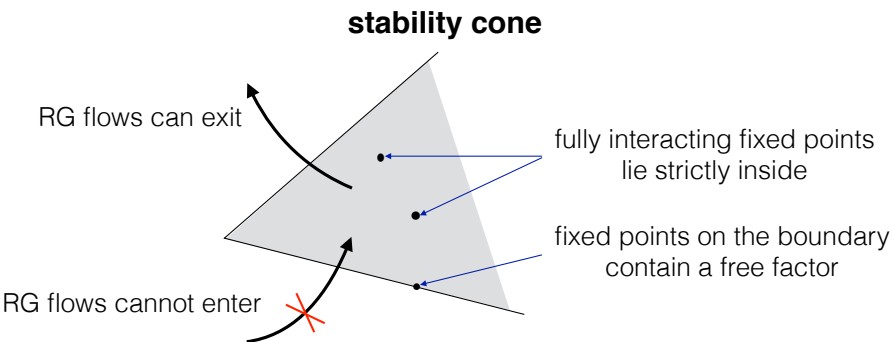

Figure 1: Graphical summary of the results of section 2.

of $\mathcal{C}$ if it starts there: RG flows cannot enter the stability cone. Second, there cannot be any fixed point in the complement of $\mathcal{C}$.[3]

Notice, however, that RG flows can *exit* the stability cone, as explicit examples show. A well known example occurs in theory of the cubic anisotropy as discussed e.g. in [6, sec. 11.3]. Such RG flows are known as "fluctuation driven first-order phase transitions".

Can there be fixed points on the boundary of the stability cone? Being on the boundary means that there is a *flat* direction $\bar{\phi}_i$ in field space, $\lambda(\bar{\phi}) = 0$, while in all other directions the potential is non-negative.

Let us introduce some terminology. The fixed point $\lambda = 0$ is called *trivial* or *free*. Two fixed point tensors $\lambda$ and $\tilde{\lambda}$ that can be transformed into one another by an $O(N)$ transformation (change of basis) of course describe physically equivalent fixed points. If we can split the fields, perhaps after a change of basis, into two subsets so that only fields within each group interact with each other, such a fixed point is called *factorized*. Finally, a fixed point which cannot be factorized is called *fully interacting*.

We will now show that all fixed points on the boundary of the stability cone are either free, or contain a free factor. This will imply that all fully interacting fixed points lie strictly inside the stability cone: 3 is strictly positive for all nonzero $\phi$; see Fig. 1.

Let $\lambda$ be a fixed point on the boundary of the stability cone, and $\bar{\phi}$ be a flat direction. Rotating fields we can assume that $\bar{\phi} = (1, 0, 0, \ldots)$ points in direction 1. Then, $\lambda(\bar{\phi}) = 0$ means that $\lambda_{1111} = 0$. We would like to show that all other couplings involving at least one index 1 vanish, so that the $\phi_1$ subsector is completely free. To do this we use the fixed point condition. Using that the beta-function for $\lambda_{1111}$ vanishes, we obtain

$$0 = \beta_{1111} = -\varepsilon \lambda_{1111} + 3\lambda_{11mn}\lambda_{11mn} = 3\lambda_{11mn}\lambda_{11mn}, \tag{7}$$

from where we conclude that all couplings $\lambda_{11mn}$ vanish. Now we use the beta-function equation for $\lambda_{11jj}$ where $j$ is an arbitrary index (no sum on $j$):

$$0 = \beta_{11jj} = -\varepsilon \lambda_{11jj} + (\lambda_{11mn}\lambda_{jjmn} + 2\lambda_{1jmn}\lambda_{1jmn}) = 2\lambda_{1jmn}\lambda_{1jmn}, \tag{8}$$

where we used that all $\lambda_{11mn}$ were already proved to vanish. Therefore, $\lambda_{1jmn} = 0$ for any $j, m, n$, which is what we need.

## 3 Review of known fixed points

In this section we will attempt a review of known fixed points. Naturally we will only mention fully interacting fixed points, as defined in section 2. A fundamental characteristic of any fixed

---

[3]This second observation can also be shown directly from the beta-function equation, which implies $\lambda_*(\phi) = 3\varepsilon B V_{ij}V_{ij} \geqslant 0$ for a fixed point $\lambda_*$ [5].

point is its symmetry group $G$, which is defined as the *maximal* subgroup of $O(N)$ that leaves the tensor $\lambda_{ijkl}$ invariant.

Notice that if $G \neq O(N)$, then by applying an $O(N)$ transformation not in $G$ the tensor $\lambda_{ijkl}$ is transformed to a different tensor $\tilde{\lambda}_{ijkl}$, which nevertheless describes the same physics. Classifying fixed points means classifying solutions of the beta-function equation $\beta_{ijkl} = 0$ up to this equivalence relation. However, there is usually one choice of field basis where the fixed point tensor $\lambda$ takes a particular simple form.

The symmetry group of any fixed point is at least as large as $\mathbb{Z}_2$, which acts by simultaneous sign flips on all fields. Curiously, all known fixed points for $N \geqslant 2$ have a strictly larger symmetry group. It would be interesting to understand why it is so.

**Open problem.**[4] Construct a fully interacting $N \geqslant 2$ scalar one-loop fixed point in $4 - \varepsilon$ dimensions with real couplings and just $\mathbb{Z}_2$ symmetry, or prove that all such fixed points have strictly larger symmetry.

An important characteristic of the symmetry group $G$ are the numbers $I_2$ and $I_4$ of quadratic and quartic invariants, $A_{ij}\phi_i\phi_j$ and $B_{ijkl}\phi_i\phi_j\phi_k\phi_l$, where $A_{ij}$ and $B_{ijkl}$ are linearly independent symmetric two- and four-tensors invariant under $G$ (to count them we choose and fix a basis in field space). When $G = O(N)$ there is just one quadratic, $\vec{\phi}^2$, and one quartic, $(\vec{\phi}^2)^2$, invariant, and the question is if there are more when the symmetry is reduced. The number of quartic invariants is clearly important since the fixed point tensor $\lambda$ will be a linear combination of $I_4$ independent invariant tensors.

While quadratic invariants do not enter directly into the analysis of RG equations, their number is important for the physical interpretation of the fixed points. Terms quadratic in fields are strongly relevant perturbations of the potential. $G$-noninvariant quadratic terms are forbidden by symmetry, while all $G$-invariant quadratic terms have to be fine-tuned to zero to reach the fixed point. Groups $G$ for which $\vec{\phi}^2$ remains a single quadratic invariant are particularly interesting, since fixed points with such symmetry would require less fine-tuning to be realized in an experiment.[5] A single quadratic invariant ($I_2 = 1$) is equivalent to requiring that the fundamental representation of $O(N)$ remains irreducible under $G$.

Historically, most attention was dedicated to fixed points with $I_2 = 1$. Notice, however, that a full classification requires considering fixed points that do not necessarily satisfy this condition.[6] We will now give some prominent examples of families of fixed points (see Table 1), known to exist for infinitely many values of $N$. Some of them have a discrete and some a continuous symmetry group. Our list is representative but far from complete; see e.g. [8,9] for more examples. We will then discuss what is known about classification.

Maximal symmetry $G = O(N)$ is realized for the **O(N) fixed point** with quartic potential given by $\lambda(\vec{\phi}^2)^2$. It exists for any $N \geqslant 1$, and reduces for $N = 1$ to the Ising (also called Wilson–Fisher) fixed point.

---

[4]A bottle of Dom Pérignon champagne will be awarded for a solution of this problem. Please contact the authors for collecting the prize.

[5]Notice that some of the quartic perturbations may also be relevant, but those require additional analysis. See section 6.

[6]Ref. [7, sec. 3] contains a remark which seems to suggest that fixed points with $I_2 > 1$ can always be factorized into a product of pairwise noninteracting fixed points with $I_2 = 1$. This cannot be correct as the example of biconical fixed point below shows. Fortunately this remark is quite tangential in [7] and does not affect the validity of the main considerations.

[7]For $m_1 = m_2 = m$, this fixed point is a particular case of the MN fixed point with $n = 2$, and the symmetry is enhanced to $O(m)^2 \rtimes \mathbb{Z}_2$, reducing the number of quadratic invariants to 1.

Table 1: Summary of examples of fully interacting fixed points given in text.

| Name | $N$ | $G$ | $I_4$ | $I_2$ |
|---|---|---|---|---|
| $O(N)$ | $N \geqslant 1$ | $O(N)$ | 1 | 1 |
| cubic | $N \geqslant 3$ | $(\mathbb{Z}_2)^N \rtimes S_N$ | 2 | 1 |
| tetrahedral | $N \geqslant 4$ | $S_{N+1} \times \mathbb{Z}_2$ | 2 | 1 |
| bifundamental | $N = mn$ | $O(m) \times O(n)/\mathbb{Z}_2$ | 2 | 1 |
| | $(m, n \geqslant 2, R_{mn} \geqslant 0)$ | | | |
| "MN" | $N = mn$ | $O(m)^n \rtimes S_n$ | 2 | 1 |
| | $(m, n \geqslant 2, m \neq 4)$ | | | |
| tetragonal | $N = 2n \geqslant 4$ | $(D_8)^n \rtimes S_n$ | 3 | 1 |
| Michel | $N = r_1 \cdots r_k$ | $G_{r_1 \dots r_k}$ | $k+1$ | 1 |
| biconical[7] | $N = m_1 + m_2$ | $O(m_1) \times O(m_2)$ | 3 | 2 |

## 3.1 Fixed points with $I_2 = 1$, $I_4 = 2$: general theory

We next consider fixed point symmetries which allow two quartic and one quadratic invariant. There is a neat general theory of such fixed points, which we will now review. First of all they satisfy the famous trace condition of [2]:

$$\lambda_{iijk} = \varepsilon z \delta_{jk}, \tag{9}$$

where $z$ is some (fixed-point dependent) constant. Indeed, the trace in the left-hand side is a $G$-invariant two-tensor and since $I_2 = 1$ it must be proportional to the tensor $\delta_{jk}$. It is then natural to write $\lambda$ as a sum of two terms:

$$\lambda_{ijkl} = \varepsilon \left( \tfrac{1}{N+2} z \, T_{ijkl} + d_{ijkl} \right), \qquad T_{ijkl} = \delta_{ij}\delta_{kl} + \delta_{ik}\delta_{jl} + \delta_{il}\delta_{jk}. \tag{10}$$

As a consequence of (9), the tensor $d_{ijkl}$ defined by this equation will be symmetric and traceless.

We now impose that the coupling (10) satisfies the beta-function equation. Using the fact that $d_{ijkl}$ is symmetric and traceless, the beta-function equation reduces to

$$d_{ijmn}d_{klmn} + d_{ikmn}d_{jlmn} + d_{ilmn}d_{jkmn} = p \, T_{ijkl} + q \, d_{ijkl}, \tag{11}$$

with coefficients $p, q$ given by

$$p = \frac{1}{N+2} z \left( 1 - \frac{N+8}{N+2} z \right), \qquad q = 1 - \frac{12}{N+2} z. \tag{12}$$

Notice that Eqs. (10), (11) and (12) followed from the trace condition only. This will be useful in section 5.4.

Now we will use the assumption $I_4 = 2$, i.e. that the space of invariant symmetric four-tensors is two-dimensional. We take as its basis elements $T_{ijkl}$ and another tensor $d^G$, chosen traceless without loss of generality.[8] Then, the tensor $d$ in (10) must be proportional to $d^G$: $d = \alpha d^G$. Notice that the tensor $d = d^G$ is bound to satisfy Eq. (11) with some $p = p_G$, $q = q_G$. Indeed, the left-hand side of (11) is a $G$-invariant symmetric four-tensor, so it must be expressible as a linear combination of $T$ and $d^G$. Notice also that (11) implies

$$d_{ijkl}d_{ijkl} = \tfrac{1}{2}pN(N+2), \tag{13}$$

---

[8]This additional tensor $d^G$ is called "primitive", because by assumption it cannot be reduced to products of lower-rank invariant tensors.

and so $p_G > 0$ since we assume that $d^G$ is not identically vanishing.

The tensor $d = \alpha d^G$ will then satisfy Eq. (11) with $p = \alpha^2 p_G$, $q = \alpha q_G$. Substituting these into (12) we see that in order to find the fixed point we must solve

$$\alpha^2 p_G = \frac{1}{N+2} z\left(1 - \frac{N+8}{N+2} z\right), \qquad \alpha q_G = 1 - \frac{12}{N+2} z, \tag{14}$$

for $\alpha$ and $z$. Since $p_G \geqslant 0$ we find $0 \leqslant z \leqslant \frac{N+2}{N+8}$.

There are two solutions of (14), $(\alpha_+, z_+)$ and $(\alpha_-, z_-)$, with

$$\alpha_\pm = \frac{1}{q_G} \frac{(N+2)\rho \pm 6\sqrt{\Delta}}{144 + (N+8)\rho}, \qquad z_\pm = \frac{(N+2)(24 + \rho \mp \sqrt{\Delta})}{2(144 + (N+8)\rho)}, \tag{15}$$

where $\rho = q_G^2/p_G \geqslant 0$ and

$$\Delta = \rho(\rho - 4(N-4)). \tag{16}$$

If $\rho \geqslant 4(N-4)$ we have a pair of fixed points with real couplings, which coincide for $\rho = 4(N-4)$. If $\rho < 4(N-4)$, these fixed points have complex couplings and are discarded since here we are interested in real fixed points.[9] A related discussion about the presence of pairs of fixed points has appeared in [2,9].[10]

Fixed points obtained using this construction are usually fully interacting, but sometimes they factorize. This happens for one of the cubic symmetry fixed points, and for one of the MN fixed points; see below.

## 3.2 Fixed points with $I_2 = 1$, $I_4 = 2$: examples

We will now consider several examples where the above general theory can be applied.

The **cubic fixed point** has $G = (\mathbb{Z}_2)^N \rtimes S_N$ symmetry, called the cubic group [9,12–14]. This is the symmetry group of the unit cube in $N$ dimensions. The second quartic invariant is $\sum_i \phi_i^4$, in the frame in which $G$ acts by permuting the fields, and by flipping their signs. Forming the $d^G$ tensor for cubic symmetry and computing the $\rho$ parameter we find

$$\rho_C = \frac{9(N-2)^2}{2(N-1)} \qquad (N \geqslant 3). \tag{17}$$

We see that $\rho_C > 4(N-4)$ for all $N \geqslant 3$, and so for any $N \geqslant 3$ there are two fixed points with this symmetry. One is fully interacting, while the other consists of $N$ decoupled copies of the Ising fixed point.

The **tetrahedral fixed points** have $G = S_{N+1} \times \mathbb{Z}_2$, the tetrahedral group [9,15]. Consider vectors $e_i^\alpha$, $\alpha = 1, \dots, N+1$ satisfying

$$\sum_\alpha e_i^\alpha = 0, \quad e_i^\alpha e_i^\beta = \delta^{\alpha\beta} - \frac{1}{N+1}, \tag{18}$$

which are vertices of the perfect hypertetrahedron in $\mathbb{R}^N$. The $S_{N+1}$ part of $G$ is the symmetry group of this hypertetrahedron, permuting the vertices, and $\mathbb{Z}_2$ acts by flipping the sign of all fields. The second quartic invariant involves the tensor $\sum_\alpha e_i^\alpha e_j^\alpha e_k^\alpha e_l^\alpha$. This is sometimes referred to as the restricted Potts model. The $\rho$ parameter for the tetrahedral symmetry equals

$$\rho_T = \frac{9(N^2 - 3N - 2)^2}{2(N-2)(N-1)(N+1)} \qquad (N \geqslant 3). \tag{19}$$

---

[9]Although in other contexts fixed points with complex couplings may be useful, see [10,11].

[10]More generally, Ref. [2] shows that given any fixed point $\lambda_*$ satisfying the trace condition, whatever its symmetry, there is another fixed point $\lambda_{**}$ which can be expressed as a linear combination of $T$ and $\lambda_*$.

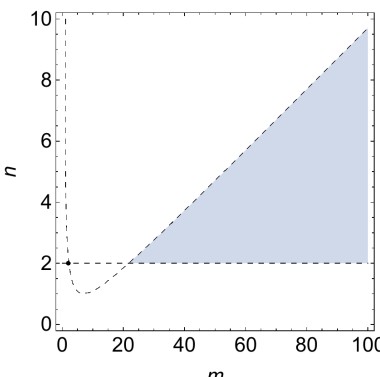

Figure 2: The region of $(m, n)$ space satisfying the conditions $m \geqslant n \geqslant 2$, $R_{mn} \geqslant 0$ needed to have bifundamental fixed points with real couplings. It consists of the point $m = n = 2$, and of the integer points in the gray region, described by Eq. (22). Notice that the allowed region around point $m = n = 2$ is really a tiny triangle invisible on this scale, but $m = n = 2$ is the only integer point within it.

We can check that $\rho_T \geqslant 4(N - 4)$ for all $N \geqslant 3$, with a strict inequality for all $N$ except for $N = 5$ where it's an equality. There are therefore two fixed points with this symmetry for $N \geqslant 4$, which coincide for $N = 5$. For $N = 3$ the tetrahedral group $G$ is isomorphic to the cubic group, and the tetrahedral fixed points coincide with the cubic ones (indeed $\rho_C = \rho_T = \frac{9}{4}$ for $N = 3$).

The **bifundamental fixed points**[11] have $G = O(m) \times O(n)/\mathbb{Z}_2$, where $N = mn$ [8,9,16,17]. We restrict to $m \geqslant n$ without loss of generality. To realize these fixed points, one writes $\phi_i$ as a $m \times n$ matrix field $\Phi_{ab}$, which transforms as a bifundamental of $O(m) \times O(n)$. The second quartic invariant is $\text{tr}(\Phi\Phi^T\Phi\Phi^T)$. Since both $O(m)$ and $O(n)$ can realize the same overall sign flip $\Phi \to -\Phi$ we need to mod out by a $\mathbb{Z}_2$.

Using results of [9] we find the $\rho$ parameter

$$\rho_{\text{bif}} = \frac{\big(mn(m+n) + 4mn - 10(m+n) - 4\big)^2}{3(m-1)(m+2)(n-1)(n+2)} = 4(N-4) + \frac{(mn+2)^2 R_{mn}}{3(m-1)(m+2)(n-1)(n+2)}, \tag{20}$$

where

$$R_{mn} = m^2 + n^2 - 10mn - 4(m+n) + 52. \tag{21}$$

To have fixed points with real couplings we need to impose $R_{mn} \geqslant 0$. Restricting to $m \geqslant n$, this is satisfied by $m = n = 2$ and

$$2 \leqslant n \leqslant 5m + 2 - 2\sqrt{6(m+2)(m-1)} \qquad (m \geqslant 22), \tag{22}$$

see Fig. 2. If $R_{mn} > 0$ there are two fixed points with real couplings (which are sometimes referred to as the chiral and antichiral fixed points), which coincide if $R_{mn} = 0$.

The Diophantine equation $R_{mn} = 0$ has an infinite number of positive integer solutions given by [18]

$$m_i = 10m_{i-1} - n_{i-1} + 4, \quad n_i = m_{i-1}, \quad i = 1, 2, \ldots, \tag{23}$$

$$m_1 = 7, \quad n_1 = 1. \tag{24}$$

---

[11]This is our proposed terminology. Sometimes they are called $O(m) \times O(n)$ fixed points.

Since $n > 1$ the solution with smallest $N$ is $m = 73$, $n = 7$, $N = 511$.

We finally consider the **MN fixed points**. They have $G = O(m)^n \rtimes S_n$, where again $N = mn$ but in this case there is no symmetry between $m$ and $n$ [7,9,19–22]. In this case $\phi_i$ is decomposed as $n \geq 2$ vectors of size $m$, $\vec{\varphi}_r, r = 1, \ldots, n$, and the second quartic invariant is $\sum_r (\vec{\varphi}_r^2)^2$. The case $m = 1$ is equivalent to the cubic, and the case $m = n = 2$ to the bifundamental symmetry.[12]

Using again results of [9], the $\rho$ parameter is given by

$$\rho_{\text{MN}} = \frac{\left(m(m+8)(n-1) + (m-4)(m+2)\right)^2}{6m(m+2)(n-1)} = 4(N-4) + \frac{(m-4)^2(mn+2)^2}{6m(m+2)(n-1)}. \tag{25}$$

General theory predicts that there are two real fixed points for $m \neq 4$, which coincide for $m = 4$. However, only one of these fixed points is fully interacting, while the other is factorized, consisting of $n$ decoupled $O(m)$ theories. For $m = 4$ only the factorized fixed point remains.

### 3.3 Further examples of fixed points

As an example of a fixed point with three quartic invariants and one quadratic invariant, we mention the **tetragonal fixed point** [9, 19, 23, 24], [6, sec. 11.6], which exists for even $N = 2n \geq 4$. The quartic potential includes the isotropic $O(N)$ term, the cubic term $\sum_i \phi_i^4$, and the tetragonal anisotropy. The latter takes the form $\phi_1^2 \phi_2^2 + \phi_3^2 \phi_4^2 + \ldots + \phi_{N-1}^2 \phi_N^2$. Focussing on the first pair of fields $\phi_1, \phi_2$, permuting them and flipping signs generates the eight-element dihedral group $D_8$. The full symmetry group of this fixed point is therefore $(D_8)^n \rtimes S_n$.

There exist fixed points with arbitrary large $I_4$. An example is provided by the **Michel fixed point** [7,9,25], which is a generalization of the MN fixed point. Consider $N = r_1 \cdots r_k$ where $r_i > 1$ are integers (not necessarily prime), and write $\phi_i$ as a tensor with $k$ indices $\alpha_i = 1 \ldots r_i$, $\Phi_{\alpha_1 \ldots \alpha_k}$. The quartic term

$$\sum_{\alpha_1 \ldots \alpha_l} \left( \sum_{\alpha_{l+1} \ldots \alpha_k} \Phi_{\alpha_1 \ldots \alpha_k}^2 \right)^2 \tag{26}$$

breaks $O(N)$ to $O(m)^n \rtimes S_n$ where $n = r_1 \cdots r_l$ and $m = r_{l+1} \cdots r_k$. The Michel fixed point contains $k + 1$ such terms ($0 \leq l \leq k$) with nonzero couplings, so that the symmetry group $G_{r_1 \ldots r_k}$ is the intersection.

Finally we mention the **biconical fixed point**, which provides an example with more than one quadratic invariant. Let us split $\vec{\phi}$ into two vectors $\vec{\phi}_1$ and $\vec{\phi}_2$ with $m_1$ and $m_2$ components, $m_1 + m_2 = N$, and consider the symmetry group $O(m_1) \times O(m_2)$ acting on $\vec{\phi}_1$ and $\vec{\phi}_2$. There are two quadratic invariants, $\vec{\phi}_1^2$ and $\vec{\phi}_2^2$, and the quartic potential is a linear combination of three invariants, $(\vec{\phi}_1^2)^2$, $(\vec{\phi}_2^2)^2$, and $\vec{\phi}_1^2 \vec{\phi}_2^2$ [26–28].

### 3.4 Classification results

Full classification of fixed points is available only for $N = 1$ and $N = 2$. Namely, for $N = 1$ we have only two fixed points, both with $G = \mathbb{Z}_2$: the free one at $\lambda = 0$ and the Wilson–Fisher fixed point at $\lambda = \varepsilon/3$.

For $N = 2$ we have only one fully interacting fixed point: the $O(2)$ one; see [9] for a completely general proof.

For $N = 3$ there are three known fully interacting fixed points: $O(3)$, cubic, and biconical. The $O(3)$ and cubic fixed points are the only ones assuming the "single quadratic invariant" condition (or a more general "isotropy constraint" $\lambda_{iklm}\lambda_{jklm} \propto \delta_{ij}$ [29]). The biconical fixed

---

[12]As a check, $\rho_{\text{MN}}$ coincides with the cubic $\rho_C$ for $m = 1$, $n \geq 2$, and with the bifundamental $\rho_{\text{bif}}$ for $m = n = 2$.

point has $O(2) \times \mathbb{Z}_2$ symmetry [26–28].[13] It would be nice to prove rigorously that there are no further fixed points.

Under the assumption of a single quadratic invariant, an extensive analysis of fixed points was performed in [30] for $N = 4$, and in [31–33] for $N = 6$. These works identified dozens of fixed points, corresponding to various discrete subgroups of $O(4)$ and $O(6)$, respectively. They offer a glimpse of the incredible complexity that a full classification of fixed points is bound to entail.

## 4 The *A*-function

As we have seen in the previous section, there are many fixed points. Ideally we would like to understand all fixed points and RG flows connecting them. There are currently only partial results towards this goal.

It has been observed long ago by Wallace and Zia [34, 35] that the one-loop beta-function can be written as a gradient of an *A*-function:

$$\beta_{ijkl} = \frac{\delta}{\delta \lambda_{ijkl}} A, \qquad A = -\tfrac{1}{2}\varepsilon\, \lambda_{ijkl}\lambda_{ijkl} + \lambda_{ijkl}\lambda_{klmn}\lambda_{mnij}\,. \tag{27}$$

We use the variational and not the usual derivative in (27) because couplings are real symmetric rank-four tensors, so the components $\lambda_{ijkl}$ are not all independent. Eq. (27) thus means that the variation of *A* is expressible as

$$\delta A = \beta_{ijkl}\,\delta\lambda_{ijkl}\,. \tag{28}$$

This is the same convention as when varying with respect to the metric in general relativity.

Equivalently we can consider a bigger vector space of all real rank-four tensors, call it $V_4$, of which the vector space of symmetric couplings, $V_4^{\mathrm{sym}}$, is a subspace. The *A*-function can be formally considered as given by the same equation on the full $V_4$. The variational derivative in (27) can be computed as the usual partial derivative $\frac{\partial}{\partial\lambda_{ijkl}}$ applied to the so-extended *A*-function.

It will also be helpful to write Eq. 27 in a form which refers to an independent set of coordinates on $V_4^{\mathrm{sym}}$. Let $\lambda^I$ be such a set of coordinates and $g_{IJ}$ be the restriction of the flat metric on $V_4$ to $V_4^{\mathrm{sym}}$.[14] Then 27 can be equivalently expressed as[15]

$$\beta^I = g^{IJ}\partial_J A\,. \tag{29}$$

Eq. (28) or its more covariant form (29) imply that the *A*-function decreases along the RG flow (flowing towards the IR). Indeed we get $(d/dt)A = \beta^I \partial_I A = g^{IJ}\partial_I A\,\partial_J A \geqslant 0$.

The existence of the *A*-function plays a fundamental role in the classification of RG fixed points and of RG flows connecting them.

One useful consequence is as follows. Take an arbitrary RG trajectory. One possibility is that the trajectory runs out to infinity. Consider the more interesting possibility that the trajectory stays bounded for all times. From a general theorem about real-analytic gradient flows due to Łojasiewicz [36–38], we can conclude:[16]

---

[13]We thank Matthijs Hogervorst for reminding us about the $N = 3$ biconical fixed point.

[14]For example we can choose as $I$ ordered tuples $ijkl$. It's easy to see that with this choice $g_{IJ} = p_I \delta_{IJ}$ where $p_I$ is the number of non-identical permutations of the tuple $I$. E.g. $p_{1111} = 1$, $p_{1112} = 4$, etc.

[15]We will write $\lambda^I$ and $\beta^I$ as required by the differential geometry conventions on contravariant and covariant indices. However, we will keep lower indices in $\lambda_{ijkl}$ and $\beta_{ijkl}$. Hopefully this will not cause confusion.

[16]Analyticity of *A* is important. For example, one can construct a $C^\infty$ gradient flow with a trajectory whose limit set is not a single point but a segment. For real-analytic *A* such pathologies are impossible.

**Fact.** Any bounded RG trajectory necessarily goes to a fixed point.

Of particular interest is the value $A_*$ of $A$ at the fixed point. Contracting the beta-function equation $\beta_{ijkl} = 0$ with $\lambda_{ijkl}$ we have

$$\varepsilon\,\lambda_{*ijkl}\lambda_{*ijkl} = 3\,\lambda_{*ijkl}\lambda_{*klmn}\lambda_{*mnij}\,, \tag{30}$$

where $\lambda_{*ijkl}$ stands for a fixed point coupling value. Using this in equation (27) for $A$ we have

$$A_* = -\tfrac{1}{6}\,\varepsilon\,\lambda_{*ijkl}\lambda_{*ijkl}\,, \tag{31}$$

from where we see that $A_*$ is always negative. It is clearly interesting to know how negative $A_*$ can become. One of the main results of our paper will be to establish a general lower bound:

$$A_* \geqslant -\tfrac{1}{48}N\,\varepsilon^3\,. \tag{32}$$

Such a bound was previously observed in [9] for a class of RG flows preserving a subgroup of $O(N)$. Here we will show that it is completely general. In particular, it holds independently of any assumption about the symmetry and the number of quadratic and quartic invariants. It applies both to fully interacting and factorized fixed points.

Equivalently, (32) says that all fixed points belong to a known compact region of coupling space:

$$\lambda_{*ijkl}\lambda_{*ijkl} \leqslant \tfrac{1}{8}N\varepsilon^2\,. \tag{33}$$

Any search of new fixed points can therefore be restricted to this region.

# 5 A bound on $A$

In this section we will prove the bound (32) on the value of $A$, or the equivalent bound (33), at any fixed point. We will omit the star subscript for fixed point values, something that will hopefully not cause any confusions. Just in this section, it will be convenient to further rescale the couplings $\lambda \to \lambda/\varepsilon$. Before rescaling the fixed point coupling is O($\varepsilon$), after rescaling it's O(1). The rescaled one-loop fixed point equation takes the form

$$\lambda_{ijkl} = \lambda_{ijmn}\lambda_{mnkl} + 2\text{ permutations}\,. \tag{34}$$

We will show that any real symmetric four-tensor solving this equation satisfies the bound

$$S = \lambda_{ijkl}\lambda_{ijkl} \leqslant C_N\,, \qquad C_N = \tfrac{1}{8}N\,. \tag{35}$$

This is equivalent to (33) after undoing the rescaling $\lambda \to \lambda/\varepsilon$.

## 5.1 Why a bound is expected to exist

First we present a simple argument which explains why a bound is expected to exist. We will fix $N$ and we will try to show that all components of $\lambda_{ijkl}$ are bounded by some constant. Like in an argument seen in section 2, the idea is to first consider components $\lambda_{iiii}$, then $\lambda_{iimn}$, and finally the general case.

For components $\lambda_{iiii}$, considering for definiteness $i = 1$, the beta-function 34 implies

$$\lambda_{1111} = 3\lambda_{11mn}\lambda_{11mn} \geqslant 3\lambda_{1111}^2\,. \tag{36}$$

From here we can conclude two things. First, since $\lambda_{1111} \geqslant 3\lambda_{1111}^2$, we must have[17]

$$0 \leqslant \lambda_{1111} \leqslant \tfrac{1}{3}. \tag{37}$$

Second, from the first equality in (36) and from (37) we have

$$\lambda_{11mn}\lambda_{11mn} = \tfrac{1}{3}\lambda_{1111} \leqslant \tfrac{1}{9}. \tag{38}$$

In particular, for any $m, n$

$$|\lambda_{11mn}| \leqslant \tfrac{1}{3}. \tag{39}$$

Finally let us bound components $\lambda_{ijkl}$ where no two indices are equal. Take for definitenes $i = 1$, $j = 2$, and impose the beta-function equation (34) for $\lambda_{1122}$:

$$\lambda_{1122} = 2\lambda_{12mn}\lambda_{12mn} + \lambda_{11mn}\lambda_{22mn}. \tag{40}$$

From here we have

$$2\lambda_{12mn}\lambda_{12mn} = \lambda_{1122} - \lambda_{11mn}\lambda_{22mn}. \tag{41}$$

The first term in the right-hand side is bounded by (39), while the second term can be bounded in absolute value using (38) and the Cauchy–Schwarz inequality:

$$|\lambda_{11mn}\lambda_{22mn}| \leqslant (\lambda_{11mn}\lambda_{11mn})^{1/2}(\lambda_{22pq}\lambda_{22pq})^{1/2} \leqslant \tfrac{1}{9}. \tag{42}$$

Summing the obtained bounds for all components, we will get a bound of the form (35) with some constant $C_N$. This reasoning is rather crude and does not give an optimal constant, in particular $C_N$ will grow quadratically with $N$ because one will have to sum over all pairs $i, j$ with $i \neq j$. In the next section we will present the argument producing $C_N = \tfrac{1}{8}N$.

## 5.2   The bound with $C_N = \tfrac{1}{8}N$

We first introduce some notation. In this section the Einstein summation convention will be applied to indices $m, n$, but all summation in indices $i, j$ will be explicitly indicated.

Denote $x_i = \lambda_{iiii}$. From (37) we know that $x_i \in [0, \tfrac{1}{3}]$. Denote also $(\mathbf{v}_i)_{mn} = \lambda_{iimn}$, viewed as matrices in $m, n$ indices. Then the first equality in (36) can be written as

$$\text{tr}\,\mathbf{v}_i^2 = \tfrac{1}{3}x_i. \tag{43}$$

The beta-function equation for the components $\lambda_{iijj}$,

$$\lambda_{iijj} = 2\lambda_{ijmn}\lambda_{ijmn} + \lambda_{iimn}\lambda_{jjmn}, \tag{44}$$

can be written as

$$\lambda_{ijmn}\lambda_{ijmn} = \tfrac{1}{2}\big((\mathbf{v}_i)_{jj} - \text{tr}(\mathbf{v}_i\mathbf{v}_j)\big). \tag{45}$$

The quantity we need to bound takes the form (using (43) and (45))

$$S = \sum_i \lambda_{iimn}\lambda_{iimn} + \sum_{i \neq j} \lambda_{ijmn}\lambda_{ijmn} = \tfrac{1}{3}\sum_i x_i + \tfrac{1}{2}\sum_{i \neq j}(\mathbf{v}_i)_{jj} - \tfrac{1}{2}\sum_{i \neq j}\text{tr}(\mathbf{v}_i\mathbf{v}_j). \tag{46}$$

---

[17]Using the notation of section 2 this can also be written as $0 \leqslant \lambda(\bar{\phi}) \leqslant \tfrac{1}{3}$ for any unit-length $\bar{\phi}$ [2].

Using further the identity

$$\sum_{i \neq j} \mathrm{tr}(\mathbf{v}_i \mathbf{v}_j) = \mathrm{tr}\Big(\sum_i \mathbf{v}_i\Big)^2 - \sum_i \mathrm{tr}\,\mathbf{v}_i^2 = \mathrm{tr}\Big(\sum_i \mathbf{v}_i\Big)^2 - \tfrac{1}{3}\sum_i x_i, \qquad (47)$$

we rewrite (46) as

$$S = \tfrac{1}{2}\sum_i x_i + \tfrac{1}{2}\sum_{i \neq j}(\mathbf{v}_i)_{jj} - \tfrac{1}{2}\mathrm{tr}\Big(\sum_i \mathbf{v}_i\Big)^2. \qquad (48)$$

We estimate the last term in the right-hand side of (48) as follows:

$$\mathrm{tr}\Big(\sum_i \mathbf{v}_i\Big)^2 = \sum_{j,k}\Big(\sum_i (\mathbf{v}_i)_{jk}\Big)^2 \geqslant \sum_j\Big(\sum_i (\mathbf{v}_i)_{jj}\Big)^2 = \sum_i\Big(\sum_j (\mathbf{v}_i)_{jj}\Big)^2, \qquad (49)$$

where in the last equality we rename $i \leftrightarrow j$ and use $(\mathbf{v}_i)_{jj} = (\mathbf{v}_j)_{ii}$. We can also separate the $i = j$ term which is $x_i$. Then (48) gives

$$S \leqslant \tfrac{1}{2}\sum_i x_i + \tfrac{1}{2}\sum_i\Big[\sum_{j:j \neq i}(\mathbf{v}_i)_{jj} - \Big(x_i + \sum_{j:j \neq i}(\mathbf{v}_i)_{jj}\Big)^2\Big] = \tfrac{1}{2}\sum_i[(x_i + y_i) - (x_i + y_i)^2], \qquad (50)$$

where we denote

$$y_i = \sum_{j:j \neq i}(\mathbf{v}_i)_{jj}. \qquad (51)$$

Finally introducing $z_i = x_i + y_i = \mathrm{tr}(\mathbf{v}_i)$, Eq. (50) takes the form

$$S \leqslant \tfrac{1}{2}\sum_i(z_i - z_i^2). \qquad (52)$$

Since $\max(z - z^2) = \tfrac{1}{4}$, attained at $z = \tfrac{1}{2}$, we finally obtain the claimed inequality:

$$S \leqslant \tfrac{1}{8}N. \qquad (53)$$

The key idea in the above proof was to combine the second (positive) and the third (negative) terms in (48), which becomes possible after estimating the negative term as in (49). If instead one were to neglect the negative term altogether, the resulting bound would have $C_N = O(N^2)$ as in the previous section, because the second positive term in (48) contains $O(N^2)$ terms.

**Remark.** A simple modification of the above argument gives a bound on $S$ for couplings whose beta-function is not zero, which may be of some interest. Namely, we have

$$S = \lambda_{ijkl}\lambda_{ijkl} \leqslant \tfrac{1}{8}N + \mathcal{B}, \qquad \mathcal{B} = \tfrac{1}{3}\sum_i \beta_{iiii} + \tfrac{1}{2}\sum_{i \neq j}\beta_{iijj}, \qquad (54)$$

where $\beta_{ijkl} = -\lambda_{ijkl} + (\lambda_{ijmn}\lambda_{mnkl} + 2 \text{ permutations})$ is the rescaled beta-function. In the proof, Eqs. (43) and (45) get extra terms $\tfrac{1}{3}\beta_{iiii}$ and $\tfrac{1}{2}\beta_{iijj}$ in the right-hand side, which sum up to $\mathcal{B}$. The remaining estimates are unaffected.

### 5.3 Improvements of the bound for $N = 1, 2, 3$

The following small modification produces a slightly improved bound for $N = 1, 2, 3$. Note that in the above argument we treated the variables $z_i$ entering the final estimate (52) as unconstrained, but in fact $z_i = x_i + y_i$ where $x_i \in [0, \frac{1}{3}]$ while $y_i$ can be bounded as

$$|y_i| \leqslant \sqrt{N-1} \Big( \sum_{j \neq i} (\mathbf{v}_i)_{jj}{}^2 \Big)^{1/2} = \sqrt{N-1} \sqrt{\tfrac{1}{3} x_i - x_i{}^2} \,, \tag{55}$$

where we have used the Cauchy–Schwarz inequality in the first step, and the second step follows from (43). So in fact (50) implies a more nuanced bound:

$$S \leqslant \tfrac{1}{2} N \varkappa_N \,, \tag{56}$$

where

$$\varkappa_N = \max_{x \in [0, \frac{1}{3}], y \in [0, \sqrt{N-1}(\frac{1}{3}x - x^2)^{1/2}]} (x + y - (x + y)^2) \,. \tag{57}$$

For $N \geqslant 4$ the maximum is attained at $z = x + y = \frac{1}{2}$. We can e.g. take $x = \frac{1}{4}$, $y = \frac{1}{4}$, and this satisfies the upper bound (55) on $y$, so $\varkappa_N = \frac{1}{4}$, and we go back to the original case of bound (53). On the other hand, for $N = 1, 2, 3$ we have

$$z_N = \max_{x \in [0, \frac{1}{3}]} (x + \sqrt{N-1}(\tfrac{1}{3} x - x^2)^{1/2}) < \tfrac{1}{2} \,, \tag{58}$$

and so

$$\varkappa_N = z_N - z_N^2 < \tfrac{1}{4} \,, \tag{59}$$

so that the new bound is stronger. We get

$$\varkappa_1 = \tfrac{2}{9} \,, \qquad \varkappa_2 \approx 0.24047 \,, \qquad \varkappa_3 \approx 0.24801 \,, \tag{60}$$

where for $N = 2, 3$ extremization is carried out numerically.

The $N = 1$ result is trivial and consistent with $S(\text{Ising}) = \frac{1}{9}$. The $N = 2$ result is not particularly interesting because the fixed points are classified; see section 3.4. Just as a sanity check, the two fixed points Ising+Ising and $O(2)$ both satisfy the bound, with $S(O(2)) = \frac{6}{25} = 0.24$ coming close to saturating it.

For $N = 3$ the bound takes the form $S \leqslant \frac{3}{2} \varkappa_3 \approx 0.372015$ and is of some interest, since the full classification has not yet been proven. Out of the known fixed points, $S(O(3)) = \frac{45}{121} \approx 0.371901$ comes closest to saturating the bound.

### 5.4 Saturation of the bound for $N \geqslant 4$

In this section we will consider the case when the bound (53), or equivalently (32), is best possible for $N \geqslant 4$. Turning this around, we will try to understand if it is possible to find fixed points that saturate (53).

Since the bound arose partly due to (49), to saturate it we need to saturate (49), which happens if and only if the following sum of off-diagonal terms vanishes:

$$\sum_i (\mathbf{v}_i)_{jk} = \sum_i \lambda_{iijk} = 0 \qquad (j \neq k) \,. \tag{61}$$

In addition, since $z_i = \text{tr}(\mathbf{v}_i)$, and maximization of $z_i - z_i^2$ happens for $z_i = \frac{1}{2}$, to saturate the bound we need

$$\text{tr}(\mathbf{v}_i) = \sum_j \lambda_{iijj} = \tfrac{1}{2} \qquad \text{(no sum over } i\text{)}. \tag{62}$$

Eqs. (61) and (62) can be summarized by saying that the bound will be saturated if and only if the fixed-point tensor satisfies

$$\sum_i \lambda_{iijk} = \tfrac{1}{2}\delta_{jk}, \tag{63}$$

a particular case with $z = \frac{1}{2}$ of the trace condition (9). As explained in section 3.1, fixed points satisfying the trace condition are described by Eq. (10) where $d_{ijkl}$ is a traceless symmetric tensor satisfying Eqs. (11), (12). Thus, fixed points saturating the bound are precisely solutions of (10)-(12) with $z = \frac{1}{2}$. This is true in full generality, e.g. without assuming anything about the symmetry of the considered fixed points. As a check, notice that these equations imply

$$\lambda_{ijkl}\lambda_{ijkl} = \tfrac{1}{2}Nz(1-z) \tag{64}$$

so that the bound is saturated if and only if $z = \frac{1}{2}$.

For $N = 4$, substituting $z = \frac{1}{2}$ into (12) we get $p = q = 0$. The obvious solution is $d_{ijkl} = 0$. Hence the $O(4)$ fixed point saturates the bound [9]. It also follows from (13) that it's the only solution.

Proceeding to $N \geqslant 5$, section 3.1 also provided a way to construct examples of fixed points satisfying (10)-(12) using symmetry groups with $I_2 = 1, I_4 = 2$. We would like to see when these examples saturate the bound. From (14), we conclude that

$$z = \tfrac{1}{2} \quad \Longleftrightarrow \quad \rho = 4(N-4). \tag{65}$$

Recall that by the general theory we have two real fixed points with $G$-symmetry if $\rho \geqslant 4(N-4)$, which coincide for $\rho = 4(N-4)$. We thus see that the bound is saturated for groups with $I_2 = 1, I_4 = 2$ if and only if the two $G$-symmetric fixed points coincide. This equivalence is not an accident: it turns out that fixed points saturating our bound for $N \neq 4$, no matter their symmetry, necessarily have a marginal perturbation. The proof of this fact is postponed to Appendix A. For now let us go over the examples from section 3.2 and recall when the coincidence happens.

We see that the cubic fixed points never saturate the bound. The tetrahedral fixed point saturates the bound if and only if $N = 5$. The bifundamental fixed point saturates the bound for $m, n$ solving the Diophantine equation $R_{mn} = 0$. There are infinitely many solutions given in Eq. (24), the first one being $N = 511 = 73 \cdot 7$. The MN fixed point can only saturate the bound for $m = 4$, when it factorizes into $O(4)$ fixed points, so that we don't get new fully interacting examples.

It would be interesting to look for more examples. We notice in this respect that a large number of $N = 6$ fixed points satisfying (63) has been reported in [33]. The RG-stable fixed points found there have $z = \frac{6}{11}$ in our notation, and so they do not saturate the bound, although they come the closest, among the fixed points reported there, to doing so.

To summarize, our analysis implies that the bound $S \leqslant \frac{1}{8}N$ is best possible for $N = 4, 5$, and for an infinite sequence of $N \geqslant 511$ obtained via (24). If one allows factorized fixed points, the bound can also be trivially saturated putting together decoupled copies of $O(4)$ and tetrahedral $N = 5$ fixed points, i.e. for all $N$ which can be represented as a linear combination $4m + 5n$

with nonnegative integers $m, n$. This covers all integers $N \geqslant 4$ except $N = 6, 7, 11$.[18] For these values of $N$, Eqs. (10)-(12) with $z = \frac{1}{2}$ can perhaps be investigated by brute force using computational algorithms of real algebraic geometry, although we have not attempted this.

## 6 RG stability

In this section we will present some general results about RG stability of fixed points, mostly relying on the work of L. Michel.

Let us remind the reader of some standard terminology. We call a fixed point RG-stable if all quartic deformations around it are marginal or irrelevant. This can be asserted by linearizing the beta-function equations,

$$\frac{d\lambda^I}{dt} = \beta^I(\lambda), \tag{66}$$

around the fixed point $\lambda_*$. RG stability means that the matrix $\Gamma^I{}_J = \partial_J \beta^I$ has all eigenvalues $\gamma \geqslant 0$ (we will see momentarily that all eigenvalues are real).

We will only study RG stability for the one-loop beta-function. Some deformations that are marginal at one loop may become relevant or irrelevant at higher loop order. This phenomenon is beyond the scope of our paper (see however section 6.3 for some related comments).

RG stability should not be confused with potential stability discussed in section 2, where we showed that all fixed points have stable potential.

Clearly the trivial fixed point $\lambda = 0$ is RG-unstable. Below we only examine nontrivial fixed points.

The $A$-function helps enormously to analyze RG-stability. Using Eq. (29) we get

$$\partial_J \beta^I = g^{IK} M_{KJ}, \quad M_{KJ} = \partial_K \partial_J A, \tag{67}$$

so the eigenvalue problem $\Gamma^I{}_J c^J = \gamma c^I$ for a generally nonsymmetric matrix $\Gamma$ is equivalent to the generalized symmetric eigenvalue problem

$$M_{IJ} c^J = \gamma g_{IJ} c^J \tag{68}$$

for the Hessian matrix $M_{IJ}$ of $A$ at the fixed point. This has two consequences. First, all eigenvalues $\gamma$ are real. Second, a fixed point is RG-stable if and only if the Hessian evaluated at that fixed point is positive semidefinite.

It will be interesting to inject an element of symmetry into the discussion of RG-stability. Let $H$ be a subgroup of $O(N)$ and consider the set of all quartic couplings $\Lambda_H$ which are $H$-invariant. The set $\Lambda_H$ is a linear subspace of all coupling tensors, and it is preserved by RG evolution.

Suppose $\lambda_* \in \Lambda_H$ is a fixed point. Notice that the symmetry group of $\lambda_*$ is at least as large as $H$ but may be strictly larger. It is interesting[19] to consider RG-stability of $\lambda_*$ with respect to perturbations belonging to $\Lambda_H$, which we will call RG-stability *within* $\Lambda_H$. By the same argument as above, this property holds if and only if $A$ restricted to $\Lambda_H$ has positive semidefinite Hessian at $\lambda_*$. The unrestricted RG stability corresponds to $H = \{1\}$, $\Lambda_H = \{$all couplings$\}$.

---

[18]For general natural numbers $a_1, a_2$ with $\gcd(a_1, a_2) = 1$, the largest integer $N$ for which there is no representation $N = m_1 a_1 + m_2 a_2$ with nonnegative integer $m_1, m_2$ is called the Frobenius number $g(a_1, a_2)$ of $a_1, a_2$. A theorem of Sylvester says that $g(a_1, a_2) = (a_1 - 1)(a_2 - 1) - 1$ [39]. In particular we have $g(4, 5) = 11$. Smaller numbers can be checked by hand.

[19]This is also physically important since the set of allowed perturbations of the microscopic Hamiltonian is often restricted by symmetry.

## 6.1 Uniqueness of RG-stable fixed point

**Theorem.** (Michel [5]) Suppose $\lambda_1, \lambda_2 \in \Lambda_H$ are two nontrivial nonidentical fixed points. Consider the value of the $A$-function at them: $A(\lambda_1)$, $A(\lambda_2)$. Then

- if $A(\lambda_1) \neq A(\lambda_2)$, then the fixed point with larger $A(\lambda)$ is RG-unstable within $\Lambda_H$ (while the other one may or may not be RG stable),

- if $A(\lambda_1) = A(\lambda_2)$, then both fixed points are RG-unstable within $\Lambda_H$.

As a consequence, there is at most one fixed point RG-stable within $\Lambda_H$.[20]

**Remark.** "Nonidentical" in the statement of the theorem means simply that $\lambda_1$ and $\lambda_2$ are unequal tensors. Nonidentical fixed points may be physically equivalent if they are related by an $O(N)$ transformation. The theorem still applies in this case. This remark will be important in the next section.

*Proof.* We give a pedagogical version of the original proof in [5]; another presentation can be found in [40], but it does not cover the case $A(\lambda_1) = A(\lambda_2)$, which is important for applications in section 6.2.

The main idea is to consider the restriction of $A$ to the two-plane within $\Lambda_H$ spanned by $\lambda_1$ and $\lambda_2$. The Hessian of restricted $A$ is evaluated by explicit computation, and the statements of the theorem follow.

To avoid getting lost in indices, let us denote for any symmetric four-tensors $u, v, w$,

$$(u, v) = u_{ijkl} v_{ijkl}, \tag{69}$$

$$(u, v, w) = u_{ijkl} v_{klmn} w_{mnij}. \tag{70}$$

Notice that $(u, v)$ and $(u, v, w)$ do not depend on the order of the arguments. Contracting the beta-function equations expressing the fact that $\lambda_1, \lambda_2$ are fixed points, we get the following auxiliary results:

$$(\lambda_i, \lambda_i, \lambda_i) = \tfrac{1}{3}\varepsilon(\lambda_i, \lambda_i), \qquad i = 1, 2, \tag{71a}$$

$$(\lambda_1, \lambda_1, \lambda_2) = (\lambda_2, \lambda_2, \lambda_1) = \tfrac{1}{3}\varepsilon(\lambda_1, \lambda_2). \tag{71b}$$

The first of these equations is (30), and the second one is a simple generalization.

In the above notation

$$A(\lambda) = -\tfrac{1}{2}\varepsilon(\lambda, \lambda) + (\lambda, \lambda, \lambda). \tag{72}$$

Using (71a), we recover (31)

$$A(\lambda_i) = -\tfrac{1}{6}\varepsilon(\lambda_i, \lambda_i), \qquad i = 1, 2. \tag{73}$$

We will assume without loss of generality that $(\lambda_2, \lambda_2) \geqslant (\lambda_1, \lambda_1)$ and will show that $\lambda_1$ is unstable.

We are interested in $A$ restricted to the two-plane spanned by $\lambda_1$ and $\lambda_2$, parametrized as

$$A(\lambda_1 + s\lambda_1 + t\lambda_2). \tag{74}$$

This is a cubic polynomial in $s, t$ and using (71) we could evaluate all coefficients. For our purposes of extracting the Hessian, we just evaluate the part quadratic in $s, t$, which comes out equal to

$$\tfrac{1}{2}\varepsilon(\lambda_1, \lambda_1)s^2 + \varepsilon(\lambda_1, \lambda_2)st + \varepsilon\left[(\lambda_1, \lambda_2) - \tfrac{1}{2}(\lambda_2, \lambda_2)\right]t^2, \tag{75}$$

---

[20]We stress that, as almost all results in this paper, this theorem is valid at one loop. Extra RG stable fixed points may appear in higher orders of the $\varepsilon$-expansion, or when this expansion is extrapolated to $\varepsilon = 1$. See [40] for a discussion.

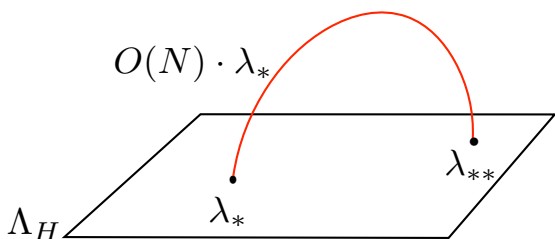

Figure 3: If the orbit $O(N) \cdot \lambda_*$ has any other intersections with $\Lambda_H$ apart from $\lambda_*$, as in this figure, the fixed point $\lambda_*$ cannot be RG-stable within $\Lambda_H$.

corresponding to the Hessian matrix

$$M = \varepsilon \begin{pmatrix} a_1 & b \\ b & 2b - a_2 \end{pmatrix}, \qquad a_i = (\lambda_i, \lambda_i), \ b = (\lambda_1, \lambda_2). \tag{76}$$

We would like to show that this Hessian is not positive semidefinite. Since the matrix element $a_1 > 0$, one of the eigenvalues is positive and we need to show that the second is negative, which will be the case if and and only if the determinant is negative. We have

$$\det M / \varepsilon^2 = 2ba_1 - a_1 a_2 - b^2 = -(a_1 - b)^2 - a_1(a_2 - a_1). \tag{77}$$

If $a_2 > a_1$, this is negative.[21] If $a_2 = a_1$, this is negative unless $b = a_1$, however the latter is impossible, since $(\lambda_1, \lambda_1) = (\lambda_2, \lambda_2) = (\lambda_1, \lambda_2)$ implies $(\lambda_1 - \lambda_2, \lambda_1 - \lambda_2) = 0$ and we are assuming $\lambda_1 \neq \lambda_2$. So in all cases $\det M$ is negative. This completes the proof that $\lambda_1$ is unstable.

## 6.2 Criteria for RG instability

To apply Michel's theorem, we need to have two fixed points. But suppose we are given only one fixed point $\lambda_* \in \Lambda_H$. We can still sometimes use the theorem to conclude that $\lambda_*$ is unstable, using an idea of [41]. Consider the orbit of $\lambda_*$ under the action of $O(N)$, denoted $O(N) \cdot \lambda_*$.

**Fact 1.** If this orbit intersects $\Lambda_H$ at some other point besides $\lambda_*$, then the fixed point $\lambda_*$ is RG-unstable within $\Lambda_H$.

*Proof.* (see Fig. 3) Let $\lambda_{**} \neq \lambda_*$ be another intersection. Since $\lambda_{**}$ is obtained from $\lambda_*$ by an $O(N)$ transformation, they have the same $A$-function: $A(\lambda_*) = A(\lambda_{**})$. By Michel's theorem, $\lambda_*$ is then RG-unstable. Of course the two fixed points are completely physically equivalent. However, $\lambda_*$ and $\lambda_{**}$ are two different *tensors*, and so Michel's theorem is applicable. QED

We thus have a sufficient condition for an RG fixed point to be unstable. As we will see now, applicability of this condition depends just on $H$ and on the symmetry group of $\lambda_*$ which we denote $G_*$. Notice that $H \subset G_* \subset O(N)$, but that $G_*$ may be strictly larger than $H$. Suppose we have

$$\lambda_{**} = g_0 \cdot \lambda_* \in \Lambda_H, \quad \lambda_{**} \neq \lambda_*, \tag{78}$$

where $g_0 \cdot$ denotes the group action of an element $g_0 \in O(N)$ on the tensor. The condition $\lambda_{**} \neq \lambda_*$ is equivalent to $g_0 \notin G_*$. On the other hand the condition $\lambda_{**} \in \Lambda_H$ is equivalent to

$$h \cdot (g_0 \cdot \lambda_*) = g_0 \cdot \lambda_* \text{ for any } h \in H \tag{79}$$

---

[21]In this case the quadratic part (75) is negative along the line $t = -s$, moving from $\lambda_1$ in the direction of $\lambda_2$ [5].

or equivalently, multiplying both sides by $g_0^{-1}$,

$$(g_0^{-1} h g_0) \cdot \lambda_* = \lambda_* \text{ for any } h \in H. \tag{80}$$

The latter condition can be expressed as

$$g_0^{-1} H g_0 \subset G_*. \tag{81}$$

We thus have an equivalent formulation of Fact 1:

**Fact 2.** Let $H \subset G_*$ be two subgroups of $O(N)$. Suppose there exists an $O(N)$ element $g_0$, such that $g_0 \notin G_*$ and $g_0^{-1} H g_0 \subset G_*$. Then any fixed point $\lambda_* \in \Lambda_H$ with symmetry $G_*$ is RG-unstable within $\Lambda_H$.

Consider now some simpler but strictly weaker conditions. Recall that the *normalizer* of any subgroup $H$ of $O(N)$ is defined as

$$N(H) = \{g : g^{-1} H g \subset H\}. \tag{82}$$

The normalizer is itself a subgroup of $O(N)$. Clearly $N(H) \supset H$ but it may be strictly larger.

Then we have

**Fact 3.** Suppose that $H \subset G_*$, and that the normalizer $N(H)$ is strictly larger than $G_*$. Then any fixed point $\lambda_* \in \Lambda_H$ with symmetry $G_*$ is RG-unstable within $\Lambda_H$.

*Proof.* This is strictly weaker than Fact 2. Take $g_0 \notin G$, $g_0 \in N(H)$. The latter implies by definition $g_0^{-1} H g_0 \subset H$ (and thus $\subset G_*$).

Specializing Fact 3 to $H = G_*$ we get:

**Fact 4.** [41] Suppose that the normalizer $N(G_*)$ is strictly larger than $G_*$. Then any fixed point $\lambda_*$ with symmetry $G_*$ is RG-unstable within $\Lambda_{G_*}$ (and thus within $\Lambda_H$ for any $H \subset G_*$).

These criteria have many applications, some of which have been explored in [41, 42]. Here we will consider only one application. Consider unrestricted RG stability: $H = \{1\}$, $\Lambda_H = \{$all couplings$\}$. By Fact 1, a fixed point $\lambda_*$ may be unrestricted RG stable only if its entire $O(N)$ orbit consists of just one point, which means that $\lambda_*$ is $O(N)$ invariant. [We can also see this from Fact 3 since $N(H) = O(N)$.]

Of course there is only one nontrivial fixed point with $O(N)$ symmetry—the $O(N)$ fixed point. It is known to be one-loop stable for $N = 2, 3, 4$,[22] while for $N > 4$ it is unstable as it flows e.g. to the cubic fixed point for these $N$. So one consequence is that for $N > 4$ there are no unrestricted RG-stable fixed points.

## 6.3 Zero RG eigenvalues: divergences of broken currents vs marginal operators

Here we would like to discuss and resolve a potential confusion related to the interpretation of zero eigenvalues of the linearized beta-function equation.

Consider a fixed point $\lambda_*$ with symmetry group $G_* \subset O(N)$. Let $G_{conn} \subset SO(N)$ be the connected component of $G_*$ containing the unity. For the free and the $O(N)$ fixed points, and only for these two, we have $G_{conn} = SO(N)$. For any other fixed point $G_{conn}$ is strictly smaller than $SO(N)$, and this is the case we wish to examine.

As usual we choose a basis of $SO(N)$ generators as $\{V_k, B_l\}$ where $V_k$ are generators of $G_{conn}$, called unbroken, while the 'broken generators' $B_l$ are a remaining set of generators completing the basis. The number of broken generators $N_B = \dim(SO(N)) - \dim(G_{conn}) > 0$ by assumption. Acting on the fixed point $\lambda_*$ with broken generators we generate a manifold $\mathcal{M}$ of tensors of dimension $N_B$. By covariance of the beta-function equation, all tensors of

---

[22]For $N = 3, 4$ the cubic deformation is marginal and at higher orders it becomes irrelevant for $N = 3$ and relevant at $N = 4$. Recall that higher order stability is beyond our scope in this paper.

$\mathcal{M}$ have zero beta-function. These are all RG fixed points, physically equivalent to $\lambda_*$ but described by different tensors.

Consider now a perturbation of $\lambda_*$ by a tensor $\delta\lambda$, in a direction tangent to $\mathcal{M}$. By the above discussion, any such perturbation will be an eigenperturbation of the linearized RG equation, with eigenvalue zero. Notice that this statement will be true to any order in perturbation theory. Does this mean we should think of such a perturbation as an exactly marginal operator? The answer is negative—these zero eigenvalues have a different interpretation.

To understand what's going on, we should consider the fate of current operators. The multiscalar theory we are studying,

$$\frac{1}{2}\partial^\mu\phi_i\,\partial_\mu\phi_i + \frac{1}{4!}\lambda_{ijkl}\phi_i\phi_j\phi_k\phi_l \tag{83}$$

has current operators

$$J_\mu = \omega_{[ij]}\phi_i\partial_\mu\phi_j, \tag{84}$$

where $\omega_{[ij]}$ parametrizes the $SO(N)$ algebra. Using the equation of motion

$$\partial^2\phi_i = \frac{1}{3!}\lambda_{ijkl}\,\phi_j\phi_k\phi_l, \tag{85}$$

the conservation equation for the current takes the form

$$\partial^\mu J_\mu = \mathcal{O}_{\delta\lambda} = (\delta\lambda)_{ijkl}\,\phi_i\phi_j\phi_k\phi_l, \qquad (\delta\lambda)_{ijkl} = \omega_{im}\lambda_{mjkl} + 3 \text{ permutations.} \tag{86}$$

At the fixed point, the currents corresponding to the unbroken generators will have $\delta\lambda = 0$ and will be conserved. On the other hand the broken generators are those for which $\delta\lambda \neq 0$. Currents corresponding to the broken generators are not conserved, and their divergence is given the quartic operators $\mathcal{O}_{\delta\lambda}$. The corresponding $\delta\lambda$'s are precisely the deformations along the manifold $\mathcal{M}$ discussed above.[23] Since $\mathcal{O}_{\delta\lambda}$ is a total derivative operator, perturbing by $\int\mathcal{O}_{\delta\lambda}$ leaves the theory unchanged. This is not the same as perturbing by an exactly marginal operator, which leads from one CFT to another, strictly different CFT.

In light of the above, it would not be even correct to ask if $\int\mathcal{O}_{\delta\lambda}$ is an irrelevant, relevant, or marginal perturbation, since it's not a perturbation at all! It still makes sense to ask what is the scaling dimension of $\mathcal{O}_{\delta\lambda}$ (as determined e.g. from the two point function), but linearized RG teaches us nothing in this respect. Indeed, the RG eigenvalue being zero means that if we add

$$g\int d^{4-\varepsilon}x\,\mathcal{O}_{\delta\lambda}(x) \tag{87}$$

to the action and perform an RG step, the coefficient $g$ does not change. However, since (87) is identically zero, adding it to the action achieves strictly nothing. That $g$ does not change contains no nontrivial information and cannot be used to draw conclusions about the scaling dimension of $\mathcal{O}_{\delta\lambda}$. To study this scaling dimension using the RG, one would need more nuanced probes, for example adding the term like (87) but with a space-dependent coupling $g(x)$; see below.

Instead, a general conclusion about the scaling dimensions of the $\mathcal{O}_{\delta\lambda}$ operators can be made by relating them to the broken currents $J_\mu$. Being broken, these currents will pick up anomalous dimensions $\gamma_J$. As usual for currents, this will first happen at two loops in the $\varepsilon$ expansion, $\gamma_J = O(\varepsilon^2)$. Importantly, these anomalous dimensions will be positive $\gamma_J > 0$ as a consequence of unitarity.[24] Their divergences $\mathcal{O}_{\delta\lambda}$ will therefore have dimensions $d + \gamma_J$ at

---

[23]Notice that any such $\delta\lambda$ satisfies the 'double tracelessness' condition $(\delta\lambda)_{iijj} = 0$. This is because the double trace is invariant under an infinitesimal $SO(N)$ transformation. This remark will be useful in Appendix A.

[24]As recently discussed in [43] the theory in $4-\varepsilon$ dimensions is not quite unitary. This absence of unitarity, however, affects only the high-dimension sector of the theory, while at low dimensions unitarity constraints still apply.

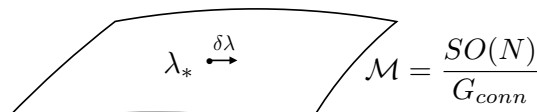

Figure 4: For fixed point perturbations by quartic scalar interactions corresponding to broken symmetry generators, linearized RG naively predicts that they are marginal. Instead such deformations correspond to total derivative operators of scaling dimension larger than $d$; see the text.

the IR fixed point. Based on their dimension, one could say that these operators are irrelevant, but as mentioned above the term 'irrelevant' is not applicable to total derivative operators. See Fig. 4.

To avoid any misunderstanding, we confirm that there is no subtlety for all the other zero eigenvalues of the linearized RG evolution, those whose eigenvectors cannot be obtained by acting on $\lambda_*$ with an $SO(N)$ generator. Their eigenvectors correspond to perturbations marginal in the one-loop approximations, which may become relevant or irrelevant in higher-loop approximation.

Let us illustrate the above discussion with a concrete example for $N = 2$. As mentioned in section 3.4, in this case there is a complete classification of fixed points [9]. We have the $O(2)$ fixed point, the free fixed point, the direct product of the Ising fixed point and a free theory, and the direct product of two Ising fixed points.

Let us focus on the latter case, which obviously does not have $O(2)$ symmetry. Let us study this theory in the frame where $\phi_1$ and $\phi_2$ are the two Ising fixed point fields (i.e. $\lambda_{1111} = \lambda_{2222}$ are the only two nonzero components of the fixed point coupling tensor). We can consider arbitrary linearized perturbations around this fixed point. The space of $N = 2$ symmetric four-tensors being five-dimensional, we have $5 \times 5$ stability matrix. It has a zero eigenvalue, corresponding to the operator $\mathcal{O}_- = \phi_1 \phi_2^3 - \phi_1^3 \phi_2$ [9]. This operator is precisely the result of acting on $\lambda_*$ with an infinitesimal rotation. Using the equations of motion we easily check that this is a descendant:

$$\mathcal{O}_- = \phi_1 \phi_2^3 - \phi_1^3 \phi_2 \sim \partial_\mu (\phi_1 \partial^\mu \phi_2 - \phi_2 \partial^\mu \phi_1). \tag{88}$$

Since $\mathcal{O}_-$ is a total derivative, a corresponding zero RG eigenvalue does not imply that its dimension is $d$. Given that we are considering a factorized theory, the dimension of the primary $\phi_1 \partial_\mu \phi_2 - \phi_2 \partial_\mu \phi_1$ is computed immediately as $d - 1 + \gamma_{\phi_1} + \gamma_{\phi_2}$, hence the dimension of $\mathcal{O}_-$ is $d + \gamma_{\phi_1} + \gamma_{\phi_2}$. This is also consistent with the fact that the dimension of $\mathcal{O}_-$ should be equal, in the factorized theory, to that of $\mathcal{O}_+ = \phi_1 \phi_2^3 + \phi_1^3 \phi_2$. Notice that since the operator $\mathcal{O}_+$ is not a total derivative its dimension is correctly predicted by linearized RG methods (it's the second eigenvector in [9, Eq. (3.12)]).

It is interesting to extract the dimension of $\mathcal{O}_-$ in a more direct way. As mentioned this can be done by performing a deformation of our theory with this total-derivative operator but with an $x$-dependent coupling. In that case, we can no longer integrate by parts to remove the deformation. Renormalization with space-dependent couplings requires new counterterms as explained in [44, 45]. The operator $J^\mu = \omega_{ij} J_{ij}^\mu = \omega_{ij} \phi_i \partial^\mu \phi_j$, $\omega_{ij} = -\omega_{ji}$ has dimension $d + \gamma_{ij}$, where the anomalous dimension is given by

$$\gamma_{ij} = (\rho_{klmn})_{ij} (\omega\lambda)_{klmn}, \qquad (\omega\lambda)_{ijkl} = \omega_{im}\lambda_{mjkl} + \text{permutations}, \tag{89}$$

with

$$(\rho_{klmn})_{ij} = (N^1_{klmn})_{ij} + \lambda_{pqrs} \frac{\partial}{\partial \lambda_{pqrs}} (N^1_{klmn})_{ij}, \tag{90}$$

where $(N^1_{klmn})_{ij}$ is the $1/\varepsilon$ pole of the counterterm $(N_{klmn})_{ij}\partial^\mu \lambda_{klmn}\phi_i\partial_\mu\phi_j$ required in the theory with space-dependent couplings. This receives contributions order by order in perturbation theory, and it has been computed that at two loops [44, 45]

$$N^1_{klmn})_{ij} = -\tfrac{1}{24}(\lambda_{iklm}\delta_{jn} - \lambda_{jklm}\delta_{in}). \tag{91}$$

From this and (90) we find for (89), at two loops,

$$\gamma_{ij} = -\tfrac{1}{12}(\lambda_{iklm}\lambda_{klmn}\omega_{jn} - \lambda_{jklm}\lambda_{klmn}\omega_{in}) - \tfrac{1}{2}\lambda_{ikmn}\lambda_{jlmn}\omega_{kl}. \tag{92}$$

In our case of two fields $\phi_1$ and $\phi_2$, where $\omega_{ij} = \epsilon_{ij}$, (92) gives, at the decoupled Ising fixed point,

$$\gamma_{12} = \tfrac{1}{12}(\lambda_{1111}^2 + \lambda_{2222}^2), \tag{93}$$

exactly as expected in order to give the dimension of $\mathcal{O}_-$ as $d + \gamma_{\phi_1} + \gamma_{\phi_2}$. Note that although we have used the fact that $\mathcal{O}_- \sim \partial_\mu(\phi_1\partial^\mu\phi_2 - \phi_2\partial^\mu\phi_1)$, the result (93) arose directly from the general expression (92), i.e. without using our knowledge of $\gamma_{\phi_1}$ and $\gamma_{\phi_2}$ in the decoupled Ising theory.

## 7 Conclusion

A very important characteristic of low loop order beta-function expressions obtained in scalar theories in $4 - \varepsilon$ dimensions is that they arise from a gradient, i.e.

$$\beta^I = g^{IJ}\partial_J A. \tag{94}$$

The main result of this paper is a general bound on the critical value of $A$ at leading loop order, given by

$$A_* \geqslant -\tfrac{1}{12}N \varkappa_N \varepsilon^3, \tag{95}$$

where $N$ is the length of the vector order parameter $\phi_i$ and

$$\varkappa_N = \begin{cases} \tfrac{2}{9} & N = 1 \\ 0.24047 & N = 2 \\ 0.24801 & N = 3 \\ \tfrac{1}{4} & N \geqslant 4 \end{cases}. \tag{96}$$

This bound demonstrates that although RG flows toward the IR cause $A$ to decrease, this cannot continue indefinitely for flows leading to a fixed point. More specifically, CFTs that are closest to saturating or actually saturate the bound (95) cannot be deformed by relevant operators and flow to other CFTs. Such deformations, if they exist, can give rise only to flows running away to large couplings and/or unstable potentials. A physical interpretation of such runaway flows is a first-order phase transition.

A perhaps more desirable way to phrase our bound would be to express it in terms of a physical quantity. The coefficient of the stress-energy tensor two-point function, $C_T$, provides us with a good candidate. At leading order there is a general result [9],

$$\frac{C_T}{C_{T,\text{scalar}}} = N - \tfrac{5}{36}\lambda_{ijkl}\lambda_{ijkl}, \tag{97}$$

where $C_{T,\text{scalar}}$ is the result for a single free scalar. Our bound can then be cast in the form

$$\frac{C_T}{C_{T,\text{scalar}}} \geqslant N\left(1 - \tfrac{5}{72}\varkappa_N\,\varepsilon^2\right). \qquad (98)$$

We have found that certain theories saturate the bound (95). Saturation of the bound can be achieved when fixed points with the same global symmetry that move in coupling space as $N$ is varied coincide. The $N = 1, 4$ cases are special—the bound is then saturated by the Ising and $O(4)$ models respectively. For $N = 2$ the bound cannot be saturated. For $N = 3$ we do not know of a theory that saturates the bound, as is also the case for theories with $N = 6, 7, 11$. For $N = 5$ the tetrahedral theory saturates the bound. Further nontrivial examples of bound saturation arise by the bifundamental fixed point, e.g. for $N = 511$. It would be interesting to compile a complete list of theories that can saturate our bound. In general, fixed points saturating the bound at $N \neq 4$ have a marginal deformation (see appendix A).

It is obviously of interest to extend our results in other directions. Within the $\varepsilon$ expansion, one could examine the fate of the bound when fermions are added. Even when Yukawa couplings are considered, it is still the case that the flow is gradient at leading order [35]. When applied to the results of [46] our bound shows that $\tilde{F}_{\text{UV}} - \tilde{F}_{\text{IR}}$ is bounded from above for scalar fixed points. It would be interesting to find a physical argument to justify this upper bound, and examine possible generalizations using the methods and results of [46].

It is also important to examine the fate of the bound beyond leading order. We remind the reader that the RG flow is gradient even at two loops in a theory with scalars and fermions in $4 - \varepsilon$ dimensions [44, 47]. In a theory with only scalars at two loops we have

$$A = -\tfrac{1}{2}\varepsilon\,\lambda_{ijkl}\lambda_{ijkl} + \lambda_{ijkl}\lambda_{klmn}\lambda_{mnij} + \tfrac{1}{12}\lambda_{ijkl}\lambda_{jklm}\lambda_{mnpq}\lambda_{npqi} - \tfrac{3}{2}\lambda_{ijkl}\lambda_{kmnp}\lambda_{lmnq}\lambda_{pqij}. \qquad (99)$$

Using the beta-function equation and the expansion $\lambda_{ijkl} = a_{ijkl}\,\varepsilon + b_{ijkl}\,\varepsilon^2 + \mathrm{O}(\varepsilon^3)$ we find, at a fixed point,

$$A_* = -\tfrac{1}{6}\varepsilon^3\,a_{ijkl}a_{ijkl} + \varepsilon^4\left(\tfrac{1}{12}a_{ijkl}a_{jklm}a_{mnpq}a_{npqi} - \tfrac{3}{2}a_{ijkl}a_{kmnp}a_{lmnq}a_{pqij}\right) + \mathrm{O}(\varepsilon^5), \qquad (100)$$

extending the result (31) beyond leading order. It is interesting that using the one-loop beta-function equation we were able to eliminate $b_{ijkl}$ from the $\varepsilon^4$ term. However, a bound pertaining to the $\varepsilon^4$ correction is not obvious. We hope to explore this possibility in future work.

In Sec. 3 we provided a review of some known fixed points in $d = 4 - \varepsilon$. For specific choices of $N$ there are many more fixed points one encounters, see e.g. [42] for $N = 4$ and [31, 33] for $N = 6$. The study of these fixed points in $d = 4 - \varepsilon$ but also in $d = 3$ with the conformal bootstrap [48] is of obvious interest and importance. As of this writing there have been only a couple of attempts in this direction [49, 50].

While the $\varepsilon$ expansion of scalar theories around four dimensions has had a long history of active research, general statements about fixed points that can be obtained within it are rather scarce. In this work we proved the bound (95), and discussed a few other general statements, some of which have appeared in the work of L. Michel. We also provided a quick review of some famous scalar fixed points. We hope that our work will provide at least an $\varepsilon$ step towards the goal of fully classifying scalar CFTs in $d = 4 - \varepsilon$ dimensions.

## Acknowledgments

We would like to thank H. Osborn for illuminating discussions and comments. We would also like to thank S. Giombi, J. Gracey, M. Hogervorst, I. Klebanov, H. Osborn and E. Vicari for

comments on the manuscript. AS would like to thank IHES and SR for hospitality. SR is supported by the Simons Foundation grant 488655 (Simons Collaboration on the Nonperturbative Bootstrap), and by Mitsubishi Heavy Industries as an ENS-MHI Chair holder. SR would like to thank the Isaac Newton Institute for Mathematical Sciences for support and hospitality during the programme "Scaling limits, rough paths, quantum field theory" when work on this paper was undertaken. This work was supported by: EPSRC grant number EP/R014604/1.

# A  Saturation of the bound and marginality

It was observed in section 5.4 that saturation of the bound on $A$ seems to go hand in hand with pairs of fixed points colliding. We saw these collisions in families of fixed points having fixed symmetry, but one may wonder if there is more general significance to this observation. The following result gives an affirmative answer:

**Fact.** For $N \neq 4$, a fixed point saturating the bound necessarily has a marginal deformation (independently of any symmetry assumptions).

The connection to fixed point collisions is obvious, since colliding fixed points are well-known to have marginal deformations. Intuitively we can understand this by using the following toy model. Suppose we have a family of RG flows continuously depending on a parameter $y$, and for each $y < y_0$ there are two fixed points that collide for $y = y_0$. Generically, close to the collision point we can focus on just one coupling $g$ whose running is described by the phenomenological beta-function

$$\beta(g) = y - y_0 + (g - g_0)^2. \tag{101}$$

The precise value of $g_0$ is not important. What is important is that for $y < y_0$ we have two fixed points at $g = g_0 \pm \sqrt{y_0 - y}$ and the dimension of the operator which couples to $g$ is $\Delta = d + \beta'(g) = d \pm 2\sqrt{y_0 - y}$ at each of them. For $y = y_0$, when fixed points collide, this operator is marginal.[25]

Let us now move past this toy model and prove the above fact. We use the general characterization of fixed points saturating the bound given in section 5.4. These are precisely fixed points that can be written in the form (10)-(12) with a symmetric traceless tensor $d_{ijkl}$ and $z = \frac{1}{2}$. For convenience we copy these conditions here:

$$\lambda_{ijkl} = \tfrac{1}{N+2} z \, T_{ijkl} + d_{ijkl}, \qquad T_{ijkl} = \delta_{ij}\delta_{kl} + \delta_{ik}\delta_{jl} + \delta_{il}\delta_{jk},$$
$$d \vee d = \tfrac{1}{3}(p\, T + q\, d), \qquad p = \tfrac{1}{N+2} z \left(1 - \tfrac{N+8}{N+2} z\right), \qquad q = 1 - \tfrac{12}{N+2} z. \tag{102}$$

Here, following [5], we introduced the vee product of two symmetric four-tensors $u, v$ which is the symmetric four tensor $u \vee v$ defined by

$$(u \vee v)_{ijkl} = \tfrac{1}{6}(u_{ijmn}v_{mnkl} + u_{ikmn}v_{mnjl} + u_{ilmn}v_{mnjk} + u \leftrightarrow v). \tag{103}$$

We are interested in $N \neq 4$ because for $N = 4$ the $O(N)$ fixed point ($d_{ijkl} = 0$) is the only solution.

We would like to find a perturbation $u$ of the fixed point that is marginal. By section 6, a marginal direction is a zero eigenvector of the Hessian $H$ of the $A$-function around the fixed point, which means

$$H_{uv} = 0 \quad \text{for any } v. \tag{104}$$

---

[25] See [10] for a more detailed review, and for what happens at $y > y_0$ when fixed points go to the complex plane.

It will be convenient to evaluate this matrix element in an index-free way, as

$$H_{uv} = \partial_s \partial_t A(\lambda + tu + sv). \tag{105}$$

Expressing the $A$-function in the notation of section 6 as (we set $\varepsilon = 1$)

$$A = -\tfrac{1}{2}(\lambda, \lambda) + (\lambda, \lambda, \lambda), \tag{106}$$

we find

$$H_{uv} = -(u, v) + 6(\lambda, u, v) = (-u + 6\,\lambda \vee u, v). \tag{107}$$

This vanishes for any $v$ if and only if

$$\lambda \vee u = \tfrac{1}{6}u, \tag{108}$$

which is therefore the condition for the existence of a marginal direction $u$.

Let us show that this equation has a solution. We will look for a solution in the form

$$u = T + x\,d \tag{109}$$

for some unknown $x$. It's easy to compute

$$T \vee T = \tfrac{N+8}{3}\,T, \qquad T \vee d = 2d, \tag{110}$$

while $d \vee d$ is given in (102). The parameter $x$ has to satisfy two linear equations. For $z \neq \tfrac{1}{2}$ one finds that there is no solution, while precisely for $z = \tfrac{1}{2}$ the two equations become linearly dependent and one finds

$$x = -\tfrac{12(N+2)}{N-4}. \tag{111}$$

Therefore, we have a marginal direction as claimed. Notice that the worries from section 6.3 do not apply: $u$ cannot be obtained from $\lambda$ acting by an $SO(N)$ generator, as it does not satisfy the double tracelessness condition, see footnote 23.

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
