# Peer review of "General Properties of Multiscalar RG Flows in $d=4-\varepsilon$"

_SciPost Physics, doi:SciPost Phys. 6, 008 (2019)_

## Round 3 · Referee Report · Anonymous (Referee 1) · 2018-11-27

Strengths

  1. Provides a clear review of classic results on scalar theories with quartic potentials that will be useful for contemporary researchers interested in studying these theories using modern techniques.

  2. Derives an interesting new bound at 1-loop order on a quantity that determines the stability of the fixed point (discussed in the Report below).

  3. The writing is clearly written and concise.

Weaknesses

  1. There are some inconsistencies in notation between section 3 and later sections regarding the factors of $\epsilon$ and $B$ in the Beta function (detailed in the Requested Changes section below).

  2. The paper could benefit from more discussion regarding which, if any, of these 1-loop results on the stability of fixed points are expected to hold non-perturbatively at the physical value $\epsilon=1$.

Report

The main result of this paper is a bound on the quantity $A$ that is related to the 1-loop Beta function in scalar theories with quartic interactions in the 4-$\epsilon$ expansion, whose symmetry group is always a subgroup of $O(N)$. A theorem of Michel shows that this $A$ is related to the stability of RG flows at this order, such that fixed points with minimal $A$ ($A$ is negative) are stable to this order.

The authors verify that the $O(N)$ fixed points for $1\leq N\leq 4$, which are known to be 1-loop stable, indeed have the minimal values $A$ of all theories that they checked, and the $O(4)$ theory in fact saturates the bound. This bound on $A$ is especially interesting for $ N>2$, where such scalar theories have not yet been classified. For $N>4$, they find that other scalar theories, such as the tetrahedral theory for $N=5$ and the bifundamental theory for $N=511$, saturate the bound.

Secondary results include: 1. Derivation that the quartic potential must be positive for fixed points and that QFTs with negative potentials cannot flow to fixed points 2. Resolution of confusion regarding zero eigenvalues of the linearized RG equations.

The paper also includes a review of known results of scalar theories with quartic potentials, collecting results in the literature that might not be known to many readers.

In the conclusion, the authors mention that similar results at 1-loop also apply to theories with fermions. To test the validity of these 1-loop results non-perturbatively, it would be useful to consider supersymmetric theories in which non-perturbative results are available.

Requested changes

  1. the Beta function in eq. 1.2 is defined in terms of $\epsilon$ and $B$. Later in section 4 $B$ is set to $1$, and in section $5$, $\lambda$ is redefined at the fixed point so that we can set $\epsilon=1$. In section 3, before these redefinitions have been announced, the authors write formulae that seem to assume such redefinitions, e.g. eq, 3.3 and 3.4. These formulae either need to include $B$ and $\epsilon$, or the redefinitions need to be said earlier.

  2. In footnote 3, $V(\lambda_*,\phi)$ is not defined, and I suspect the RHS should include a factor of B.

---

## Round 3 · Referee Report · Anonymous (Referee 2) · 2018-12-22

Strengths

1- The paper is written in a very clear language and is a pleasure to read. 2- The paper proves an interesting new lower bound on an RG height-function. 3- The paper serves as a useful review of the existing knowledge on an important class of RG fixed points.

Weaknesses

1- The article does not seem to discuss the effects of higher-loop corrections on the conclusions.

Report

This paper is a step forward in the important but very hard problem of classifying conformal field theories. It focuses on the workable yet still very rich class of fixed points of scalar theories in the epsilon expansion.

While the article does not seem to discuss effects of higher-loop corrections on the conclusions, such discussion is bound to be rather complex, given the already rather high complexity of the one-loop story. Still, in my opinion some discussion of which conclusions should stay valid at higher orders, or finite value of $\epsilon$ should be attempted.

Requested changes

1- if possible, add a remark on expected effects of higher-loop corrections

---

## Round 4 · Author Response

We would like to thank the referees for the careful reading of our manuscript. As suggested by Report 1, we rescaled B away earlier in the manuscript, and fixed footnote 3. We also added a few comments about next-to-leading order results in the conclusion, specifically around eq. (7.7), and footnote 20.

---

## Editorial Decision

published